# *Think Yourself Slim?* Assessing the Satiation Efficacy of Imagined Eating

**DOI:** 10.3390/foods12010036

**Published:** 2022-12-22

**Authors:** Tjark Andersen, Derek Victor Byrne, Qian Janice Wang

**Affiliations:** 1Food Quality Perception & Society, Department of Food Science, Faculty of Technical Sciences, Aarhus University, Agro Food Park 48, DK-8200 Aarhus N, Denmark; 2Sino-Danish College (SDC), University of Chinese Academy of Sciences, Beijing 100049, China

**Keywords:** grounded cognition, sensory-specific satiety, food intake, priming, mental imagery

## Abstract

Ubiquitous exposure to visual food content has been implicated in the development of obesity with both individual and societal costs. The development and increasing adoption of Extended Reality (XR) experiences, which deliver an unprecedented immersion in digital content, would seem to carry the risk of further exacerbating the consequences of visual food exposure on real-world eating behavior. However, some studies have also identified potentially health-promoting effects of exposure to visual food stimuli. One example is repeated imagined consumption, which has been demonstrated to decrease subsequent food consumption. This work contains the first comparison between imagined eating and actual eating, to investigate how the simulated activity fares against its real counterpart in terms of inducing satiation. Three-hundred participants took part in an experiment at a local food festival. The participants were randomized between three experimental conditions: imagined eating, actual eating, and control. Each condition consisted of thirty trials. Before and after the experimental manipulation, the participants recorded their eating desires and enjoyment of a piece of chocolate candy. The resulting data showed generally no difference between the imagined eating and control conditions, which stands in conflict with the prior literature. In contrast, the differences between imagined and actual eating were significant. These results may be explained by differences in the experimental tasks’ dose–response relationships, as well as environmental-contextual disturbances. Overall, the findings do not corroborate the efficacy of imagined eating within a real-life context.

## 1. Introduction

People are constantly exposed to visual food content—whether on television or online [1,2]. Systematic reviews and meta-analyses have found clear connections between visual food content exposure, e.g., through advertisement, and adverse eating behavioral outcomes, such as increased total food intake and weight gain [3,4]. Such “priming” activates respective mental representations, causing top-down alterations in attention, desire, and, ultimately, behavior [5,6,7]. In particular, food images suggestive of high energy density seem to be most readily attention-capturing [8,9]. This is problematic, as obesity imposes significant welfare costs on individuals and society [10,11]. 

Digital food exposure and the associated appetite stimulation can be expected to increase further. Trends indicate that people are steadily spending more time on digital and online experiences [12,13]. At the same time, the technological development of “Extended Reality” (XR) applications will make for ever more immersive experiences. For example, the company Meta (formerly Facebook), whose business relies on advertisements, is committing large investments to the development of its Metaverse [14,15]. This context urges the continued scientific exploration of the effects of digital food exposure and advertisement, as well as the development of countermeasures against any adverse health influences.

Opportunely, emerging research is already demonstrating that visual food stimuli do not necessarily and exclusively lead to bad health outcomes. For example, two studies by Brede et al. [16,17] showed that viewing food pictures for 8 min can reduce the subsequent blood glucose response after an ad-libitum meal or oral glucose tolerance test. Other research suggests that food images could facilitate satiation. Larson et al. [18] found that the enjoyment of tasting peanuts was reduced after evaluating sixty salty food pictures, compared to sixty sweet pictures or only twenty salty or sweet pictures. Similarly, repeated mentally imagined eating can satiate and lead to a subsequent reduction in food intake [19,20,21]. For instance, one study that also controlled for total cognitive load showed that imagining eating a food thirty times reduced the intake of that food compared to zero or three times ([21], replicated by [22]).

However, the abovementioned satiation literature leaves at least two aspects undetermined. First, all studies record outcome measures only after the experimental manipulation and, therefore, lack a comparative baseline. Thus, it is unclear whether group differences are attributable to satiation or priming. Second, studies have compared imagined eating only to a cognitively equivalent control. No study to date compared the satiation effects of imagined and actual eating. In the analogous case of priming, an experiment by Lambert et al. [23] found looking at a food picture to be as effective as tasting the food. It is, therefore, intriguing to investigate whether imagined eating is also as capable of inducing satiation as actual eating. This comparison would help to theoretically estimate the potency of imagery-induced satiation and, to an extent, even mental imagery interventions in general.

## 2. Aim and Hypotheses

This work aimed to investigate how effectively imagined eating satiates, in comparison to a control task and actual eating. Here, it was important to also differentiate priming and satiation. The following two hypotheses formalize the expected outcomes:

**H1.** *Imagined eating is more satiating than a cognitive control task*.

**H2.** *Imagined eating is just as satiating as actual eating*.

Figure 1 schematizes the hypothesized relationship.

## 3. Materials and Methods

### 3.1. Recruitment

The study was conducted at a popular food festival in Denmark. Participants were recruited at the university department’s tent, between approximately 11:00 and 19:00 local time. The recruitment target was 300 participants, to be equally divided between three conditions (described in Section 3.2). This sample target is similar to the replication of Morewedge et al. [21] by Camerer et al. [22], who recruited 80 participants per group. Experimenters recruited participants for one condition at a time. To ensure randomisation, the experimental condition was changed every hour (based on a cycle of three conditions). Participants were required to be at least 18 years of age and to review and electronically provide informed consent before taking part in the study. Due to the anonymous data collection and the commonplace nature of the study, formal ethics approval was not required. Table 1 shows the participants’ demographic data after the removal of outliers (described in Section 3.3).

### 3.2. Procedure

Participants were seated at café tables and equipped with an iPad running the study as well, as a small 2 cl plastic glass holding two orange M&Ms (Mars, Inc., McLean, VA, USA). The iPad contained all procedural instructions (participants in the actual eating group received further oral instruction to pay attention to the written prompt to raise their hand), which were available in both English and Danish. After giving their informed consent, participants evaluated their current hunger level, eating desire (desire-to-eat, sweet, salty, and fatty; [24]), and general liking of M&Ms [18]. Hereafter, participants tasted and subsequently rated their enjoyment of an orange M&M (cf. [18]).

For the experimental manipulation, subjects went through 30 trials of either: (1) actual eating, (2) imagined eating, or (3) a control task.

**Condition 1:** In the actual eating task, participants raised their hands and were subsequently given a small zip-lock bag with 30 M&Ms of five different colors. Orange M&Ms were removed to avoid exposure effects vis-à-vis the earlier orange “enjoyment” M&M. The task of this group was to simply eat the M&Ms, one by one, as guided by the iPad study. 

**Condition 2:** The imagined eating group saw 30 pictures of the same M&Ms as in the actual eating group, individually plated on a ramekin. The pictures were taken under standardized conditions (see Figure 2, panel A). During the task, participants were asked to “vividly imagine eating” the depicted M&M. 

**Condition 3:** The control group saw 30 pictures of five colored tokens of the same colors as the M&M stimuli (for lack of a brown token, a purple token was used instead). To approximate the contrast of the imagined eating stimuli, the tokens were “plated” on a white rounded rectangular shape (see Figure 2, panel B). Participants were asked to imagine inserting the depicted token into a laundry machine. This task has previously been used as a cognitive load control for imagined eating [20,21].

To avoid participants casually skipping through the study, all groups implemented an obligatory two-second delay for each “trial”, after which, participants could advance to the next one at their own pace.

After the experimental manipulation, participants rated their eating desires again. Hereafter, the participants ate the second orange M&M and rated their enjoyment of it. As a final experimental measure, participants saw a menu of one through ten plated orange M&M’s ((see Figure 2, panel C) and were asked to select the number of M&M’s they felt like eating at the moment (cf. [19,21]). The experiment ended with the collection of basic demographic data (age and gender), as well as a free-text field to gather participants’ feedback on their study experience. Figure 3 visualizes the study procedure.

### 3.3. Data Analysis

Data analysis was conducted in *R* v4.1.0 (R Core Team, R Foundation for Statistical Computing, Vienna, Austria.). Participants’ time spent on each trial was covertly recorded and used for outlier removal. It was assumed that unusually fast or slow trial times were signs of lacking engagement. Outliers were defined as individuals’ median trial time below $Q_{1}-1.5\cdot IQR$ or above $Q_{3}+1.5\cdot IQR$ (*IQR*: Inter Quartile Range; $Q_{1}$: First quartile, i.e., 25’th percentile; $Q_{3}$: Third quartile, i.e, 75’th percentile. $IQR=Q_{3}-Q_{1}$). In total, 15 outliers were removed (*n_imagined eating_* = 7, *n_control_* = 8).

The general approach to significance testing was to construct statistical models at the highest complexity (e.g., with interacting dependent variables) and then step-wise reduce the complexity via likelihood-ratio tests. The significance level was set at *α* = 0.05. Further, all models controlled for the general liking of M&Ms (except for the comparison of general liking, described below), hunger, age and gender. These covariates were not statistically investigated, as they were not of theoretical interest. 

As a control of the randomization procedure, initial general liking of M&Ms was compared between groups using a simple linear model. Repeated measures data (i.e., enjoyment of M&Ms and eating desires) were modelled with linear mixed models (LMM) with random intercepts for participants and tested for group differences and within-subject time (pre- and post-experiment) differences. The desired number of M&Ms was tested for post-experimental manipulation group differences with a generalized linear model (GLM) with the Poisson link function.

The final models then underwent post hoc testing of either planned or pairwise contrasts, as indicated in the results. Planned contrasts were between-group comparisons at “pre” and “post” time points, as well as “pre-to-post” within-group comparisons, for a total of nine comparisons. Adjustment for multiple comparisons was based either on the Tukey or Bonferroni correction, as appropriate. Finally, effect sizes were estimated as Cohen’s *d*.

## 4. Results

The different groups’ participants spent dissimilar amounts of time on each trial (actual eating: median = 9.0 s, IQR = 6.0 s; imagined eating: 3.4 s, IQR = 2.1 s; control: median = 3.1 s, IQR = 1.2 s). The trial time in the actual eating group steadily decreased over the experiment, whereas it remained constant for the imagined eating and control groups. The IQR values are reflective of this dynamic. There was no difference in general liking of M&Ms between the groups (*F*_(2,294)_ = 0.855, *p* = 0.43).

The desire to eat something was significantly influenced by the interaction of time and group (χ^2^_(2)_ = 18.669, *p* < 0.001). Post hoc tests adjusted for nine comparisons showed that all groups had similar initial levels (all *p* = 1). After the experiment, there was no difference between the imagined eating and control groups (*p* = 1). The actual eating group had a significantly lower desire than the imagined eating (CE = −9.680 ± 2.70, *d* = 0.61, *t*_(353)_ = −3.585, *p* = 0.003) and control groups (CE = −13.855 ± 2.82, *d* = 0.87, *t*_(557)_ = −4.916, *p* < 0.001). All groups saw significant within-subject decreases, i.e., from before to after the experimental manipulation (actual eating: CE = −20.707 ± 2.26, *d* = 1.31, *t*_(303)_ = −9.151, *p* < 0.001; imagined eating: CE = −9.811 ± 2.14, *d* = 0.62, *t*_(303)_ = −4.591, *p* < 0.001; control: CE = −7.544 ± 2.37, *d* = 0.48, *t*_(303)_ = −3.179, *p* = 0.015).

The desire for something sweet was significantly affected by the interaction of time and group (χ^2^_(2)_ = 56.764, *p* < 0.001). All post hoc tests were Bonferroni adjusted for nine comparisons. There were no initial differences between the groups (all *p* ≥ 0.35). After the experiment, the imagined eating and control groups were no different from each other (*p* = 0.24). The actual eating group differed from both the imagined eating (CE = −20.81 ± 3.19, *d* = 1.13, t_(544)_ = −6.522, *p* < 0.001) and control groups (CE = −24.61 ± 3.33, *d* = 1.34, t_(549)_ = −7.394, *p* < 0.001). Only the actual eating group saw a significant pre-to-post decrease (CE = −25.98 ± 2.62, *d* = 1.41, t_(303)_ = −9.904, *p* < 0.001; other *p* ≥ 0.15).

The desire for something salty was significantly affected only by the main effect of time (χ^2^_(1)_ = 24.653, *p* < 0.001), decreasing pre-to-post (CE = −6.26 ± 1.24, *d* = 0.42).

The desire for something fatty was significantly influenced by the interaction of time and group (χ^2^_(2)_ = 13.649, *p* = 0.001). Post hoc tests were Bonferroni adjusted for nine comparisons. All groups were initially similar (all *p* ≥ 0.62). Additionally, after the experiment, there were no statistically significant group differences (all *p* ≥ 0.06). Both the actual eating (CE = −9.26 ± 1.50, *d* = 0.89, t_(303)_ = −6.196, *p* < 0.001) and imagined eating groups (CE = -4.37 ± 1.41, *d* = 0.42, t_(303)_ = −3.095, *p* = 0.02) saw significant pre-to-post decreases, whereas the control group saw no change (*p* = 1).

The enjoyment of M&Ms was significantly affected by the interaction of time and group (χ^2^_(2)_ = 60.465, *p* < 0.001). Post hoc tests with Bonferroni adjustment for nine comparisons showed no initial group differences (all *p* = 1). There was also no difference between imagined eating and control after the experimental manipulation (*p* = 1). However, the actual eating group enjoyed the M&M significantly less than both the imagined eating (contrast estimate [CE] = −22.270 ± 3.10, *d* = 1.32, *t*_(519)_ = −7.174, *p* < 0.001) and control groups (CE = −24.342 ± 3.24, *d* = 1.44, *t*_(524)_ = −7.519, *p* < 0.001). All groups saw significant decreases in their enjoyment of M&Ms (actual eating: CE = −32.808 ± 2.41, *d* = 1.95, *t*_(303)_ = −13.618, *p* < 0.001; imagined eating: CE = −8.991 ± 2.28, *d* = 0.53, *t*_(303)_ = −3.952, *p* = 0.001; control: CE = −8.656 ± 2.53, *d* = 0.51, *t*_(303)_ = −3.426, *p* = 0.006).

Groups differed significantly in the desired number of M&Ms (Deviance_(2)_ = 124.65, *p* < 0.001). Post hoc pairwise comparison with Tukey adjustment revealed that the actual eating group desired significantly fewer M&Ms than the imagined eating (contrast ratio (CR) = 0.46 ± 0.03, *d* = 0.56, *z*-ratio = −10.366, *p* < 0.001) and control groups (CR = 0.519 ± 0.04, *d* = 0.47, *z*-ratio = −8.221, *p* < 0.001). The imagined eating and control groups showed no difference (*p* = 0.11).

Figure 4 visualizes the results.

## 5. Discussion

### 5.1. Imagined Eating vs. Control

The experiment showed, in general, no difference between the imagined eating and control groups. Hence, H1 has to be rejected. This was unexpected in light of the prior literature that found differences between imagined eating and control groups, even using a similar number of trials [20,21,22]. One reason may be that participants in the imagined eating group did not vividly visualize the eating experience. Unfortunately, we failed to measure mental imagery vividness (see, e.g., [25]). Trial times of the imagined eating and control groups indicate that cognitive load was similar, thus validating the choice of control task. However, these trial times were much shorter than those of the actual eating group. This might imply that, indeed, participants did not vividly visualize the eating experience. Corroborating this conjecture is the finding that trial times longitudinally decreased in the actual eating group—presumably tracking the onset of satiation—whereas they remained constant in the other two groups. However, the imagined eating group’s trial times were very comparable to corresponding groups in a set of recent online studies with approximately 1200 participants, which did find evidence for satiation [26]. These online studies share the same methodological blueprint as the present study; therefore, we could compare findings between the two studies.

Another reason may be that the festival environment was too distracting or may otherwise have undermined mental imagery attempts and the subsequent development of satiation. Missbach et al. [20] found that mental imagery-induced satiation was contingent on the availability of self-regulatory resources. The highly stimulating environment of the food festival may have depleted those resources and, hence, left participants unable to properly imagine the eating task. Moreover, measuring mental imagery vividness would have provided valuable clues. If this turned out to be the case, it could be questioned how useful such mental imagery would be in practice, i.e., outside of well-controlled lab conditions.

### 5.2. Imagined Eating vs. Actual Eating

Not only were trial times much shorter for imagined compared with actual eating, but on the majority of outcomes, satiation was much smaller too. In particular, the desire to eat, the desire for something sweet, and the enjoyment of M&Ms were all significantly higher after imagined eating as compared to actual eating (*p* < 0.001 for all comparisons). This suggests that imagined eating did not prove to be equally satiating as actual eating. H2, therefore, also has to be rejected. Excepted from this pattern was the desire for something salty or fatty, for which there were no group differences at all before or after the experiment. These were “unexposed” tastes and, from the perspective of sensory-specific satiety [27], expected to be rather unresponsive to the experimental manipulations. It follows then, that responses should be similar across conditions, which was the case indeed. The contrast between the “exposed” and “unexposed” tastes is most apparent in the actual eating group, with a considerably larger effect size for the change in desire for something sweet compared to fatty.

On the whole, the finding that imagined eating is less satiating than actual eating is plausible. However, considering the lack of differences between the imagined eating and control groups, the experiment may not accurately represent the theoretical difference in satiation between imagined and actual eating.

### 5.3. Priming vs. Satiation

Regarding the differentiation of priming and satiation, the results contained no instance of priming. All groups and outcomes showed indications of satiation, except for the desire for something sweet in the imagined eating and control groups, and the desire for something fatty in the control group. Yet, the lack of a decrease in the desire for something sweet after imagined eating need not necessarily imply that no satiation occurred. The already mentioned online studies—the results of which were congruent with the responses in the imagined eating group—found the same lack of decrease in these eating desires, yet further indicated that this was a return to baseline from a primed state [26]. It is, therefore, conceivable that the correspondence of the imagined eating and control groups is coincidental in the sense that the underlying dose–response relationships differ (see Figure 5 for an illustration). The online studies, thus, showed that imagined eating first primes (e.g., after three trials) and then satiates (e.g., after thirty trials). Such a relationship is limited to “exposed” food and tastes (i.e., in this study the enjoyment and desired quantity of M&M’s, the desire to eat in general, and the desire for something sweet in specific), and theoretically substantiated [28]. Though the dose–response relationship of the control task is unknown, a pattern of initial priming followed by satiation appears unlikely.

The study is limited by the fact that all groups were only put through thirty trials, rather than a range of trial numbers. Hence, the study by itself is unable to answer dose–response-related questions. Beyond filling this gap, future studies should measure mental imagery vividness to estimate and hypothetically distinguish between participants’ study task compliance; their innate mental imagery ability, which may also influence the result [29]; and situational impacts on mental imagery. Related to the latter point is the need to test imagined eating in other environments. If, as discussed, the lively festival environment of this study was too distracting, it may be worthwhile to explore the situational boundary conditions at which mental imagery becomes infeasible.

It has been mentioned that up-and-coming XR applications, where digital food may be presented in realistic, immersive contexts, can potentially exacerbate the issue of appetite stimulation. However, these applications are also well-suited to address the limitations of the present study. For example, realistic environmental distractions could be administered to varying degrees to study the aforementioned situational boundary of the feasibility of mental imagery. This would not only provide fine-grained experimental control, but also increase ecological validity compared to regulatory resource-depletion studies, such as the one by Missbach et al. [20].

Another research area of relevance to the design of XR applications is the effect of individual mental imagery ability on the consequences of imagined eating. Prior work by Krishna et al. [29] showed that behavioral consequences of imagining food smells differed between participants based on their mental imagery ability. In XR, it would be possible to study the effect of mental imagery ability within participants, namely by making the simulated eating experience more (or less) explicit, thereby gradually reducing (or increasing) the demand on participants’ mental imagery ability. The literature already contains analogous examples of such applications [30,31] (Example videos can be found at: http://www.okajima-lab.ynu.ac.jp/demos.html; accessed on 19 December 2022). Broadly speaking, such an application which uses technology to enhance the vividness of food imagery could be considered an “accessibility technology” for a cognitive, rather than physical, ability.

In conclusion, this work compared imagined eating with actual eating and a not-eating-related control task. The study also differentiated priming and satiation responses. However, the study yielded unexpected results that, in part, stand in contradiction to prior literature. Specifically, the study did not find imagined eating leading to satiation when compared to a control task. Imagined eating was also much less satiating than actual eating. Though the study contains limitations that could be addressed in future studies employing XR technology, its results do not support imagined eating as an effective satiation strategy in a real-life context.

## Figures and Tables

**Figure 1 foods-12-00036-f001:**
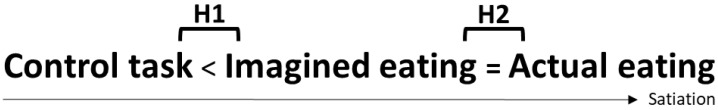
Hypotheses relationship. The figure illustrates how satiation increases from a control task to imagined eating and actual eating. H1 and H2 refer to, respectively, hypotheses 1 and 2, tested in this work.

**Figure 2 foods-12-00036-f002:**
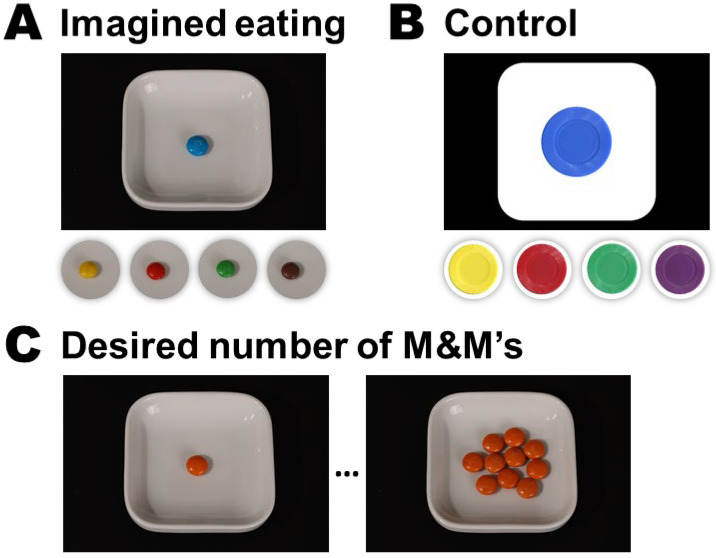
The visual stimuli used in the study. **Panels A** and **B** show the visual stimuli used during the experimental manipulation in the imagined eating **A** and control **B** groups. The actual eating group saw only a text instruction (see Figure 3); thus, it is not shown here. **Panel C** shows the first (1) and last (10) choice options when participants were asked to indicate the number of M&Ms they desired to eat at the moment, at the end of the study.

**Figure 3 foods-12-00036-f003:**
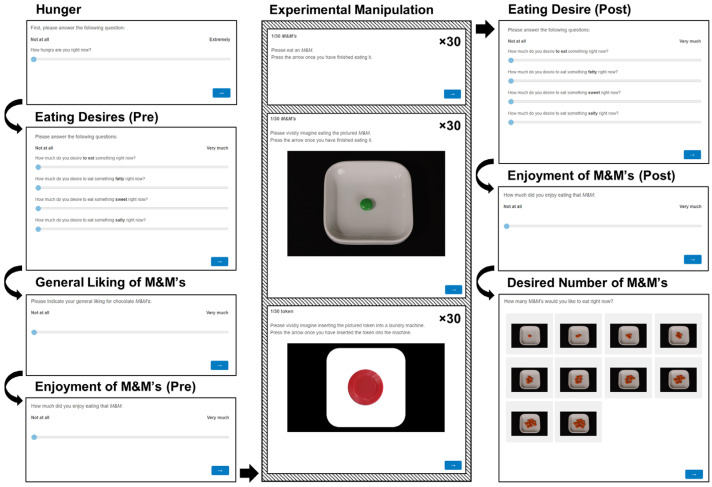
Visualization of the study procedure. The middle column, Experimental Manipulation, illustrates the tasks that the different groups were randomized to do, each repeated thirty times.

**Figure 4 foods-12-00036-f004:**
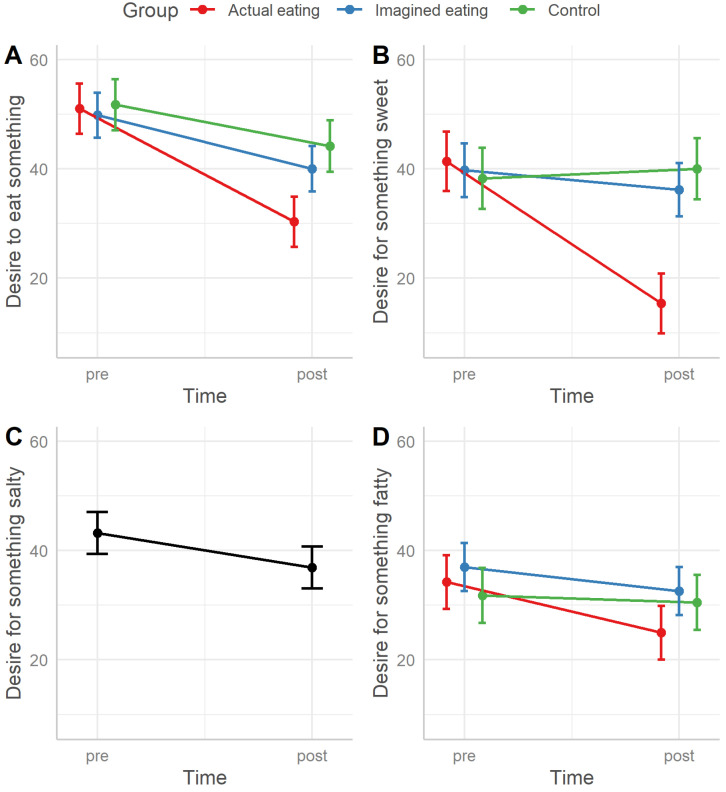
Statistical model predictions. **Panels A–D** show participants’ eating desires before (pre) and after (post) experiment. Furthermore, **panels A**, **B**, and **D** illustrate group differences, whereas the groups in panel **C** have been pooled due to no differences. **Panel E** shows the enjoyment of a tasted orange M&M before (pre) and after (post) the experiment between the groups. **Panel F** shows the desired number of M&Ms between the groups after the experiment.

**Figure 5 foods-12-00036-f005:**
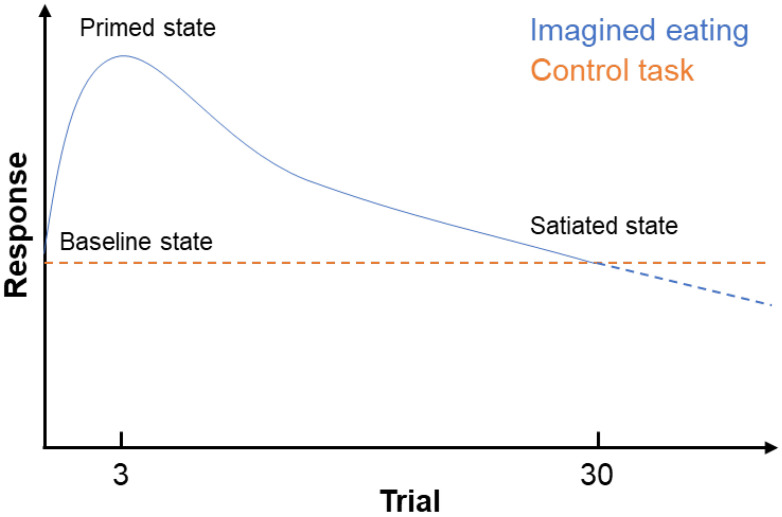
Hypothetical dose–response relationships of the imagined eating and control tasks. The curves are substantially different, yet cross around 30 trials, i.e., the experimental observation. Knowledge of the primed state stems from prior online studies (currently under review). The figure is adapted from Crolic and Janiszewski [28].

**Table 1 foods-12-00036-t001:** Demography of included participants. The aim was to recruit 100 participants per condition for a total of 300 participants.

CONDITION	N_PARTICIPANTS_	%_FEMALE_	AGE (µ ± SE)
**Actual eating**	99	68%	37.0 ± 1.4
**Imagined eating**	111	57%	33.7 ± 1.3
**Control**	90	67%	35.8 ± 1.6
	**300**	**63%**	**35.4 ± 0.8**

## Data Availability

The data presented in this study are available on request from the corresponding author.

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
