# Peer review of "Think Yourself Slim? Assessing the Satiation Efficacy of Imagined Eating"

_foods, 2022, doi:10.3390/foods12010036_

Round 1

Reviewer 1 Report

Abstract L19-20 "Each group repeated  their respective task thirty times". In methodology it is written that  "Experimenters recruited participants for an hour at a time for each condition, rotating between the three 88 experimental conditions.". Consistency necesarry

L.92-99 Remove text as it is not related to methodology "Research manuscripts reporting large datasets that are deposited in a publicly available database should specify where the data have been deposited and provide the relevant accession numbers. If the accession numbers have not yet been obtained at the time of submission, please state that they will be provided during review. They must be provided  prior to publication

L245 More explanation about study or project number or something else should be added to understand why it is important to compare with these data 

L248 necessary to add some data of statistical analysis to prove why hypothesis was rejected

Author Response

We thank the reviewer for the constructive and attentive feedback. We have incorporate the requested in the revised the manuscript. Please see the point-by-point response below:

Abstract L19-20 "Each group repeated  their respective task thirty times". In methodology it is written that  "Experimenters recruited participants for an hour at a time for each condition, rotating between the three 88 experimental conditions.". Consistency necesarry

  • We have clarified the intended meaning at the indicated sections, reproduced below:
  • Abstract: “The participants were randomised between three experimental conditions: imagined eating, actual eating, and control. Each condition consisted of thirty trials. Before and after the experimental manipulation, the participants recorded their eating desires and enjoyment of a piece of chocolate candy.”
  • Section 2.1: “Experimenters recruited participants for one condition at a time. To ensure randomisation, the experimental condition was changed every hour (based on a cycle of three conditions).”

L.92-99 Remove text as it is not related to methodology "Research manuscripts reporting large datasets that are deposited in a publicly available database should specify where the data have been deposited and provide the relevant accession numbers. If the accession numbers have not yet been obtained at the time of submission, please state that they will be provided during review. They must be provided  prior to publication

  • Section was removed (originating from the Foods manuscript template).

L245 More explanation about study or project number or something else should be added to understand why it is important to compare with these data 

  • We have added a sentence to elaborate on the relation between the study at hand and the referred online studies.
  • Section 4.1: “And yet, the imagined eating group’s trial times were very comparable to corresponding groups in a set of recent online studies with approximately 1200 participants (currently under re-view), which did find evidence for satiation. These online studies share the same methodological blueprint as the present study, therefore we could compare findings between the two studies.”

L248 necessary to add some data of statistical analysis to prove why hypothesis was rejected

  • We have added a sentence to specify which exact statistical outcome the hypothesis rejection is based on. Please see the change in its context, reproduced below:
  • Section 4.2: “Not only were trial times much shorter for imagined compared with actual eating, but on the majority of outcomes, satiation was much smaller, too. In particular, the desire to eat, the desire for something sweet, and the enjoyment of M&M's were all significantly higher after imagined eating compared to actual eating (p < .001 for all comparisons). This suggests that imagined eating did not prove to be equally satiating as actual eating. H2, therefore, also has to be rejected.”

Reviewer 2 Report

Dear Authors,

I read with great interest your article. 

It was well-written and presented, good Introduction, proper Methodology and study design, exciting Hypotheses to follow, clear  Results with significant content, strong statistics, good citation and recent and proper References. I appreciated also the limits you underline.

Based on the quality of presentation, the novelty of this virtual-nutritional intervention and assessment, strong statistics and scientific soundness, with excited interest for the readers, conclusion supported by the results, easy to be followed again and continued, in another study and a very good overall merit. 

Author Response

We thank the reviewer for the kind feedback. No requested changes.